# A Consent Support Resource with Benefits and Harms of Vaccination Does Not Increase Hesitancy in Parents—An Acceptability Study

**DOI:** 10.3390/vaccines8030500

**Published:** 2020-09-02

**Authors:** Ciara McDonald, Julie Leask, Nina Chad, Margie Danchin, Judith Fethney, Lyndal Trevena

**Affiliations:** 1Faculty of Medicine and Health, School of Medicine, University of Sydney, Sydney 2006, Australia; ciara.mcdonald@sydney.edu.au; 2ASK NHMRC Centre for Research Excellence, University of Sydney, Sydney 2006, Australia; julie.leask@sydney.edu.au; 3Faculty of Medicine and Health, Susan Wakil School of Nursing and Midwifery, University of Sydney, Sydney 2006, Australia; judith.fethney@sydney.edu.au; 4Faculty of Medicine and Health, School of Public Health, University of Sydney, Sydney 2006, Australia; nina.berry@sydney.edu.au; 5Department of General Medicine, Department of Pediatrics, The Royal Children’s Hospital, Victoria 3052 Australia; margie.danchin@rch.org.au; 6Vaccine Uptake Group, Murdoch Children’s Research Institute Victoria, University of Melbourne, Victoria 3052, Australia

**Keywords:** childhood vaccination, consent, vaccine hesitancy, information, informed choice, consent support resource

## Abstract

It is unclear whether information given about the benefits and risks of routine childhood vaccination during consent may cue parental vaccine hesitancy. Parents were surveyed before and after reading vaccine consent information at a public expo event in Sydney, Australia. We measured vaccine hesitancy with Parent Attitudes about Childhood Vaccine Short Scale (PACV-SS), informed decision-making with Informed Subscale of the Decisional Conflict Scale (DCS-IS), items from Stage of Decision Making, Positive Attitude Assessment, Vaccine Safety and Side Effect Concern, and Vaccine Communication Framework (VCF) tools. Overall, 416 parents showed no change in vaccine hesitancy (mean PACV-SS score pre = 1.97, post = 1.94; diff = −0.02 95% CI −0.10 to 0.15) but were more informed (mean DCS-IS score pre = 29.05, post = 7.41; diff = −21.63 95% CI −24.17 to −18.56), were more positive towards vaccination (pre = 43.8% post = 50.4%; diff = 6.5% 95% CI 3.0% to 10.0%), less concerned about vaccine safety (pre = 28.5%, post = 23.0%, diff = −5.6% 95% CI −2.3% to −8.8%) and side effects (pre = 37.0%, post = 29.0%, diff = −8.0% 95% CI −4.0% to −12.0%) with no change in stage of decision-making or intention to vaccinate. Providing information about the benefits and risks of routine childhood vaccination increases parents’ informed decision-making without increasing vaccine hesitancy.

## 1. Introduction

Each year, millions of children are vaccinated, diseases are prevented and lives are saved [1,2]. An important part of this is the process of establishing valid consent. Establishing valid consent to treatment is part of the ethical and legal responsibility of health workers [3]. However, there is some concern that providing parents with too much information, especially about risks of vaccination, might provoke anxiety, cue vaccine hesitancy, or discourage uptake [4]. Equally, effective routine communication about vaccination is important for preventing and managing hesitancy—a “motivational state of being conflicted about, or opposed to, getting vaccinated” [5] that may lead to a “delay in the acceptance of the vaccine despite the availability of vaccination services” [6]. The World Health Organisation (WHO) identified vaccine hesitancy as one of the top ten threats to public health globally in 2019 [7]. Despite this urgency, there has been limited assessment of the relationship between providing detailed consent information and vaccine hesitancy amongst parents.

A systematic review of parents’ views and experiences of vaccine communication (mainly in high-income countries) found that parents want more information than they are currently receiving, and that this should include both the benefits and risks of vaccination. Parents often find it challenging to locate information they can trust, that they perceive to be unbiased and balanced [8]. Some parents report regretting their decision to vaccinate, or worry about it, especially when they are not able to access the information they need [5]. Healthcare providers are a trusted source of information about vaccines for parents [9] and often influence their vaccination decisions [5]. Negative clinic experiences may even reduce the willingness of parents to return for further vaccines for their children [10]. Similar results have also been found in low and middle-income countries [11].

Studies of parents’ attitudes to childhood vaccination consistently show that the vast majority are supportive and intend to vaccinate their children [9,12,13]. Even so, around half state they have some concerns about vaccines. In particular, parents report concerns about possible side effects and vaccine safety and would like more information [9,12]. A qualitative study found that Australian parents with an “accepting” attitude to vaccination want to be involved in the decision to immunise their children. These parents reported wanting good quality information to be available to them because it indicated transparency and engendered trust [14]. Some of the parents reported that they would avoid reading information about serious side effects as it might make them more anxious but valued having it made available to them. Parents also reported that consent processes were variable and often absent. Importantly, the parents who participated in this study welcome the opportunity or an invitation to ask questions regarding their concerns [14].

There has been a lot of emphasis on understanding the drivers of vaccine hesitancy amongst parents, particularly amongst the relatively small percentage who delay or refuse to immunise their children [5,15]. However, very little is known about the effects of consent processes on the vast majority of parents who seek vaccination but may have questions or concerns about vaccine safety. This large group of parents repeatedly report that they want to be involved in the decision to vaccinate their children and to have access to balanced information about benefits and risks of vaccination. Many parents seek information online and find material of variable quality [16]; those who obtain their information from clinicians are less likely to have residual vaccine concerns [17]. Trust and good communication between providers and parents’ consultations is key to establishing valid consent for routine childhood vaccination [18]. However, clinicians are understandably cautious about providing too much information about rare or serious side effects of vaccines, lest they inadvertently alarm parents or dissuade them from vaccinating their children [9,19]. There is some evidence that adults considering their own vaccination may be more likely to trust their clinicians if information about adverse event reporting was given [20]. However, there has been very little evaluation of the impact of providing information about benefits and risks associated with routine childhood vaccination on parents who intend to vaccinate their children [21].

This study aimed to investigate whether providing parents with information about the benefits and risks of routine childhood vaccination provokes or increases vaccine hesitancy. The resource was designed to support informed consent and had the secondary aim of assessing whether parents felt that this was the case.

## 2. Materials and Methods

### 2.1. Study Design

Interventions to enhance provider–patient communication are often complex [22]. Guidance on the development and evaluation of complex interventions emphasises the importance of acceptability testing of an intervention before larger efficacy studies and implementation takes place [23,24]. This study therefore specifically focusses on acceptability testing of a consent support resource with parents. It is based on the premise that any substantial increase in vaccine hesitancy arising from parents reading the resource would be unacceptable [5]. We used a pre-post study design to investigate whether providing parents with written information about the risks and benefits of vaccination affected vaccine hesitancy (Parent Attitude About Childhood Vaccination Short Scale PACV-SS score) [25] or decision process (Decision Conflict Scale—Informed Subscale DCS-IS score) [26]. Secondary outcomes included change in scores on Stage of Decision Making, Positive Attitude Assessment [27], Side effect and Safety Concerns [25,28], and the Vaccine Communication Framework [27,29].

### 2.2. Ethics

This study was approved by the University of Sydney Human Research Ethics Committee (Number 2018/258).

### 2.3. Intervention

Development of the (Sharing Knowledge About Immunisation (SKAI)) consent support resource is reported elsewhere [14,19]. The resource is part of an Australian Government-funded digital platform designed to support parent-provider communication about routine childhood vaccination [30]. SKAI was co-designed with parents and providers by a multidisciplinary research team and is premised on a previously published vaccine communication framework [31].

The SKAI consent resource is a three-page fact-sheet, titled “What vaccines are recommended for my baby from six weeks?” that presents information about childhood vaccination using a question/answer format. It is divided into the following sections: “What vaccines are recommended to protect my baby from 6 weeks?”, “How will the vaccines affect my baby?”, “What can I do if my child gets one of these reactions?”, “Do vaccines work?”, “What are the diseases these vaccines protect my baby from?”, “Are the diseases serious?”, “I’ve heard vaccines can have serious side effects. Is this true?”, “Where can I get more information?” and “What is next?”. It is one of eight consent support resources included on the SKAI platform. Each of these contains information about the vaccines recommended at each age-based immunisation milestone and provided to children in Australia under the National Immunisation Program. These eight resources are structured consistently, addressing the key questions identified in the development phase and providing information required to meet a minimum standard for consent [14,19].

### 2.4. Recruitment and Data Collection

We recruited a convenience sample of parents from the “Parents, Babies and Children’s Expo” in Sydney in May 2018. The Expo is an annual three-day exhibition of goods and services relevant for pregnancy and early childhood attended by thousands of parents each year. The researchers paid for a booth in the exhibition space, branded with the University of Sydney logo and promoted as a survey study about childhood vaccination. Members of the research team invited parents who approached the booth to complete the online survey using the Qualtrics™ platform accessed via tablet computers provided or via a link we emailed on request. Parents were eligible for the study if they were Australian residents who could read and respond in English, were expecting a baby, or were the parent of a child less than 6 months of age.

### 2.5. Sample Size

Sample size was calculated using Raosoft [32]. Assuming a 5% margin of error, 95% confidence and target population size of 50,000, a total sample size of 382 was required. Accounting for 10% loss to follow-up and incomplete responses increased the sample size to 424.

### 2.6. Survey Items and Outcome Measures

The SKAI consent resource was embedded in the survey instrument. Participants were required to open each section (question subheading) of the SKAI consent resource before they could move to the next. This enabled us to measure time spent on each section as an indication of engagement (see Figure 1).

We collected baseline demographic, pregnancy and vaccine information from each participant before administering the pre/post-test, including age, gender, level of education, postcode, pregnancy status, gestational age, number of children, birth and six-week vaccination status of participant’s youngest child (if applicable). We also measured attitude to vaccination using a single-item measure [29] (“Overall, how do you feel about childhood vaccination?” Strongly oppose = 1, Generally oppose, Neither support nor oppose, Generally support, Strongly support = 5) and previous vaccine experiences and behaviours [25,33] (“Have any of your children experienced an adverse event after immunisation?”; “Do you know anyone who has had a bad reaction to a vaccine?”, “Have you ever delayed having a child of yours get a vaccine for reasons other than illness or allergy?”, and “Have you ever decided not to have a child of yours get a vaccine for reasons other than illness or allergy?” (Yes/No)).

The scales we used to measure primary and secondary outcomes are described in Table 1. The Parent Attitudes about Childhood Vaccines Short Scale (PACV-SS) [25] measures vaccine hesitancy and the Informed Subscale of the Decisional Conflict Scale (DCS-IS) [34] measures whether respondents feel adequately informed to make a vaccination decision. Responses to the five-item PACV-SS are scored 0, 1, or 2 and summed. Higher scores indicate higher levels of vaccine hesitancy. Responses to the DCS-IS items are scored 0, 2, or 4, summed and converted to a percentage. Higher scores indicate the respondent feels less informed.

We measured the strength and nature of parents’ vaccination decision-making using The Stage of Decision Making Scale [35]. We also included a Positive Attitude Assessment [27], a measure of Side effect and Safety [25,28], and an item designed to indicate parents’ position within the Vaccine Communication Framework (VCF) [27,29]. We included an additional question that provided parents an opportunity to indicate whether reading the consent resource had raised any concerns for them.

### 2.7. Analysis

Data were analysed using SPSS Statistics v24. (IBM Corp. Released 2016. IBM SPSS Statistics for Windows, Version 24.0. Armonk, NY: IBM Corp). For the PACV-SS and DCS-IS scores, before and after responses were compared using paired t-test analyses. We considered Safety Concerns and Side 

Effect Concerns to be dichotomous variables and used McNemar’s test to compare the proportion of responses indicating concern before and after exposure to the intervention (see Table 1).

Subgroup analyses of PACV-SS and DCS-IS responses were performed. Parents were grouped based on responses to the VCF question at baseline into two groups. The “Accepting” group included parents who responded that they would give all vaccines and had no concerns or a few minor concerns about the safety of those vaccines. The “Not Accepting” group included those in the other three categories (vaccinate with a lot of concerns, delayed or selectively vaccinate or refuse all vaccines). Parents were also grouped based on whether they reported that they had concerns raised by reading the consent support resource. We also completed a sub-group analysis which excluded those parents who did not ‘click on’ and open any of the sections of consent resource. Paired t-test analyses were again used to assess for any change in mean PACV-SS or DCS-IS scores after the intervention in parents who were not accepting at baseline or had concerns raised during the survey. A *p* value < 0.05 was considered statistically significant.

## 3. Results

### 3.1. Baseline Characteristics

In all, 487 eligible parents agreed to participate in the study. Of these, 416 parents completed the instrument and 71 did not complete the primary outcome measure (PACV-SS). There was no significant difference in age, gender, education status or attitude to vaccination between those who completed the study and those who did not (Table 2). The mean age of the sample was 31.6 years. Most respondents were female, university educated, and lived in regions of relative socio-economic advantage. Most participants were pregnant with their first child and in their second or third trimester. The majority (95.4%) were supportive of vaccination at a rate similar to estimates from the general parent population [9]. A small proportion of parents had experienced an adverse event with one of their children (3.1%) and some had delayed (3.1%) or not vaccinated (1%) for reasons other than illness or allergy

### 3.2. Outcomes

#### 3.2.1. Intervention Fidelity

All eight sections of the resource were viewed by at least half of the participants (41.8%) with 21.6% not opening any sections (See Figure 2) despite all parents reporting that they had viewed every section. Participants spent more time reviewing the SKAI consent resource if they completed it at home (mean = 217.8 ± 24.9 s) than those who completed it at the expo (mean = 61.9 ± 3.1 s). That is a mean difference of 155.9 s (95% CI 125.3 to 186.6 s).

#### 3.2.2. Primary Outcome: Hesitancy and Informed Decision Making

There was no change in mean vaccine hesitancy (PACV-SS score) which remained generally low (1.97 pre and 1.94 post, *p* = 0.71). However, there was a statistically significant reduction in the mean Decisional Conflict Informed subscale (29.05 pre to 7.41 post, *p* < 0.0001), indicating a substantial increase in proportion of parents who felt they could make an informed vaccination decision (Table 3).

#### 3.2.3. Secondary Outcomes: Stages of Decision Making, Vaccine Hesitancy Category, Attitude to Vaccination and Concerns about Side Effects and Safety

Most participants indicated that they had already made a decision about vaccinating their child at baseline and there was no significant change after the intervention (75.4% pre and 72.5% post, *p* = 0.1). Most parents were also categorised at baseline as “Accepting” on the Vaccine Communication Framework and this also remained unchanged (90.1% pre and 92.0% post, *p* = 0.1) after exposure to the intervention. There was, however, a small but statistically significant improvement in the proportion of parents with a positive attitude to vaccination (43.8% pre and 50.4% post, *p* = 0.001) and a reduction in concerns about vaccine side effects (37.0% pre and 29.0% post, *p* < 0.001) and concerns about vaccine safety (28.5% pre and 23.0% post, *p* < 0.001) (Table 4) The majority of parents stated that the resource had not raised any concerns for them (91.1%). Amongst the 34 (8.2%) respondents who indicated they had some concerns, six related to vaccine safety or side effects, two related to distrust of the Government and the information provided, two felt they had been given too much information, one wanted more information, one reported that they wanted to be advised by their doctor before vaccinating their child, and one was concerned about new vaccines not being available for free.

### 3.3. Subgroup Analyses

In the subgroup analysis of participants whose PACV-SS scores indicated they were “Not accepting” of vaccination at baseline and or who reported that reading the SKAI consent resource raised concerns, there was no significant change in their PACV-SS scores after the intervention. However, DCS-IS scores decreased amongst both sub-groups, indicating they felt more informed. Similarly, when we excluded those parents who had not “clicked on” and opened any sections of the consent resource we found similar mean PACV-SS and DCS-IS scores, a non-significant but larger reduction in hesitancy and a greater and also significant decrease in the mean DCS-IS (Table 3).

## 4. Discussion

This study found that a currently used Australian vaccine consent support resource containing information about the benefits and risks of childhood vaccination did not increase vaccine hesitancy or provoke concerns amongst the parents we sampled. This group of predominantly “accepting” parent participants felt more informed about routine childhood vaccination, less concerned about vaccine safety or potential side effects, and had a more positive attitude to vaccination after receiving the resource. These effects were sustained amongst the subgroups of parents who were “non-accepting” at baseline and those who indicated the intervention had raised some concerns for them.

Our vaccine consent support resource was deliberately designed using a rigorous development process to address the expressed needs of parents and primary care providers for more detailed information about the benefits and risks of childhood vaccination [14,15,19]. We believe it is the first to assess whether such a resource might induce greater vaccine hesitancy amongst “accepting” parents and we found that it did not. It has been argued that engagement, transparency, and parent-centred approaches are key to addressing the global threat of vaccine hesitancy [7,36].

This acceptability study used a before-and-after method, based on the premise that the consent support resource, once implemented, would form one component of a complex intervention. As such, the Medical Research Council’s guidance on development and evaluation of complex interventions was utilised [23]. As the acceptability of this vaccine consent support resource has been demonstrated, this component can be integrated confidently within the broader SKAI communication package, that also includes conversation tips for providers [37] and further evaluated in the clinical setting [31].

Further strengths of this study include the large sample size and high completion rate amongst parents who represent the target audience for this intervention. Most participants were first time mothers in their second or third trimester, which is an appropriate group for delivery of vaccination information [27]. Our study population was more advantaged than the general population, based on the distribution of Index of Relative Socioeconomic Advantage and Disadvantage (IRSAD) scores. We recruited at a commercial event that required paid entry. However, socioeconomic advantage correlates with higher rates of vaccine hesitancy and refusal in some studies, so this is an appropriate setting in which to determine whether our intervention might increase concerns in that population [38]. Under-vaccination amongst lower SES parents is more likely due to access or practical barriers to vaccination, although vaccine hesitancy is still present [39].

This study could not determine the effect of this intervention on parents who do not speak English because they were excluded from the sample. The lack of a control group limited the capacity for our study to measure magnitude of the effect of exposure to the intervention on parents’ concerns and decision-making.

All participants who completed the survey reported that they had read the SKAI information provided. While 41.8% opened each section, most (%) participants spent less than 1 min looking at the information, and 21.7% did not open any sections of the information. This suggests that much of the information was not read in detail. Parents who spent more time reading the consent resource did so at home, suggesting that these resources should ideally be provided to parents in advance of the vaccination consultation to maximise engagement. This aspect emphasised (as part of the SKAI package) the need to embed the consent resource as a more routine part of the consent process and as the standard of care for parents.

As the majority of participants had minimal concerns about vaccination, they may not have felt it necessary to read all the information in detail. This is consistent with earlier qualitative research with “accepting” parents who stated that they wanted access to more detailed information about vaccination even though some of them would not actually read it [14]. We anticipate that, once implemented, this resource would enable and empower parents to address their vaccine concerns, having built trust through the availability of a balanced consent support resource.

## 5. Conclusions

It is safe to provide parents with detailed information about the benefits and risks of routine childhood vaccination because it does not appear to increase hesitancy. Parents who are provided with this information feel more informed about their vaccination decision. Thus, providing parents with detailed information about the benefits and risks of routine childhood vaccination may prevent vaccine hesitancy. Further research is required to investigate this question.

## Figures and Tables

**Figure 1 vaccines-08-00500-f001:**
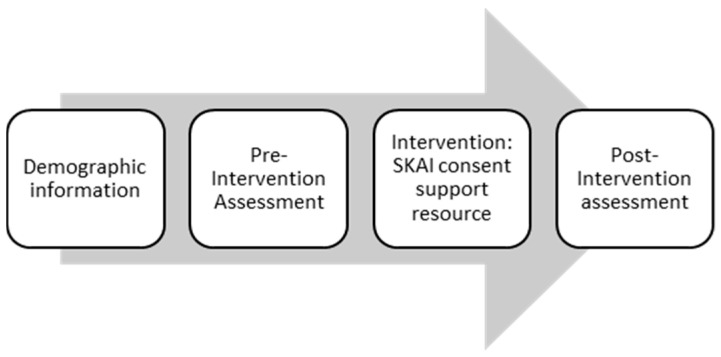
Survey Structure.

**Figure 2 vaccines-08-00500-f002:**
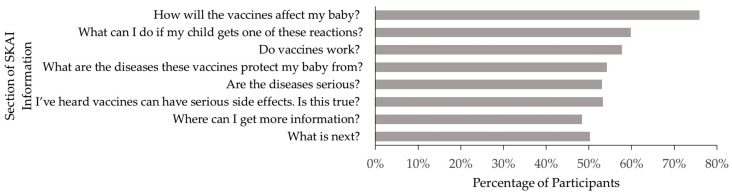
Percentage of Participants Who Viewed the Sharing Knowledge About Immunisation (SKAI) Information, by Section.

**Table 1 vaccines-08-00500-t001:** Survey Items for Pre and Post Intervention Measures.

Survey Item	Grouping/Score
Stage of Decision MakingAt this Current Time, Regarding the Decision about Vaccinating Your Baby, Would You Say You:	
haven’t begun to think about the choices	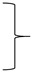	Not firmly decided
haven’t begun to think about the choices, but am interested in doing so
are considering the options now
are close to selecting an option
have already made a decision, but am willing to reconsider
have already made a decision and am unlikely to change my mind	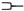	Firmly decided
Decisional Conflict Scale–Informed Subscale (DCS-IS)When thinking about vaccinating your baby: (Answer Yes, Unsure, No)Do you know which options are available to you?Do you know the benefits of each option?Do you know the risks and side effects of each option?Parental Attitude to Childhood Vaccines Short Scale (PACV-SS)(Answer Agree, Unsure, Disagree)I trust the information I receive about vaccinationsIt is better for my child to develop immunity by getting sick than to get a vaccineIt is better for my child to get fewer vaccines at the same timeChildren get more vaccines than are good for them(Answer Not hesitant, Unsure, Hesitant)Overall, how hesitant about childhood vaccines would you consider yourself to be?Positive Attitude AssessmentI would give my child any new vaccine available, even if it was not on the National Immunisation Schedule	
Strongly agree	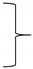	Agree
Agree
Somewhat agree
Neither agree nor disagree
Somewhat disagree	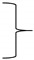	Disagree
Disagree
Strongly disagree
Side-effect ConcernsHow concerned are you that your child might have a serious side effect from a vaccine?	
Very concerned	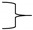	Concerned
Somewhat concerned
Not sure	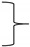	Not concerned
Not too concerned
Not at all concerned
Safety ConcernsHow concerned are you that any one of the childhood vaccines might not be safe?	
Very concerned	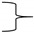	Concerned
Somewhat concerned
Not sure	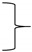	Not concerned
Not too concerned
Not at all concerned
Vaccine Communication Framework (VCF) QuestionWhich one of the following statements best applies to you at this moment?	
I will allow my baby to have all of their recommended vaccines and have NO concerns about the safety of those vaccines	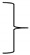	Accepting
I will allow my baby to have all of their recommended vaccines and have A FEW MINOR concerns about their safety
I will allow my baby to have all of their recommended vaccines but have A LOT OF concerns about their safety	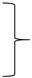	Not accepting
I will allow my baby to have some vaccines on time but there are some I prefer to delay or not have at all
I will not allow my baby to have any vaccines
I am unsure about vaccinating my baby

**Table 2 vaccines-08-00500-t002:** Baseline Characteristics of Participants (*N* = 416).

Characteristic	*N* (%)
Age (mean ± SD)	31.6 (±4.8) years
Gender (Female)	321 (77.2)
Level of Education	
Postgraduate	85 (20.4)
Bachelor Degree	183 (44.0)
Diploma or Certificate	82 (19.7)
Secondary School	60 (14.4)
Index of Relative Socio-economic Advantage and Disadvantage (IRSAD) Ranking in Deciles, By Postcode (10 represents most advantaged)	
10	116 (27.9)
9	105 (25.2)
8	52 (12.5)
7	26 (6.3)
6	26 (6.3)
5	21 (5.0)
4	27 (6.5)
3	17 (4.1)
2	10 (2.4)
1	14 (3.4)
Currently pregnant	
Yes	353 (84.9)
First Trimester	30 (7.2)
Second Trimester	170 (40.9)
Third Trimester	153 (36.8)
Have other children? (Yes)	152 (36.5)
Youngest child has received birth vaccines? (Yes)	62/152 (40.8)
Attitude towards vaccination	
Strongly oppose	2 (0.5)
Generally oppose	6 (1.4)
Neither oppose nor support	11 (2.6)
Generally support	57 (13.7)
Strongly support	340 (81.7)
Previous childhood vaccination experiences	
Own child experienced adverse event	13 (3.1)
Knows someone with adverse event	73 (17.5)
Delayed vaccination (other than illness or allergy)	13 (3.1)
Not vaccinated (other than illness or allergy)	4 (1.0)

**Table 3 vaccines-08-00500-t003:** Comparison of Hesitancy and Informed Decision Making Pre and Post Intervention.

Survey Item	Mean (SE) Pre	Mean (SE) Post	Difference	95% CI		*p*-Value
				Lower Limit	Upper Limit	
**PACV-SS Score (Hesitancy Measure) ^a^**				**-**	**-**	**-**
All participants	1.97 (0.10)	1.94 (0.11)	−0.02	−0.10	0.15	0.71
Subgroup–VCF ^c^ “Not accepting”	5.31 (0.44)	4.84 (0.45)	−0.47	−1.15	1.09	0.13
Subgroup–Concerns raised	3.82 (0.43)	4.26 (0.50)	0.44	0.24	−1.12	0.20
Subgroup-Non-viewers excluded	1.96 (0.12)	1.87 (0.13)	−0.09	−0.63	0.23	0.26
**DCS-IS Score (Informed Decision Making) ^b^**				**-**	**-**	**-**
All participants	29.05 (1.73)	7.41 (0.90)	−21.63	−24.71	−18.56	<0.0001 *
Subgroup–VCF “Not accepting”	50.88 (5.62)	15.79 (3.77)	−35.09	−46.75	−23.42	<0.0001 *
Subgroup–Concerns raised	47.06 (6.15)	14.22 (3.86)	−32.84	−45.69	−20.00	<0.0001 *
Subgroup–Non-viewers excluded	30.83 (1.96)	6.39 (0.91)	−24.44	−28.80	−20.80	<0.0001 *

^a^ PACV-SS is scored out of 10. A lower score indicates lower hesitancy. ^b^ The DCS-IS is scored out of 100. A lower score indicates that participants feel more informed. ^c^ VCF = Vaccine Communication Framework category. * Statistically significant.

**Table 4 vaccines-08-00500-t004:** Comparison of Responses to Likert-Type Items Pre and Post Intervention.

Survey Item ^a^	Response Group ^a^	Pre	Post	Difference	95% CI		*p*-Value
					Lower Limit	Upper Limit	
Positive Attitude Assessment	Agree	43.8%	50.4%	6.5%	3.0%	10.0%	0.001 *
Side-effect Concerns	Concerned	37.0%	29.0%	−8.0%	−4.0%	−12.0%	<0.001 *
Safety Concerns	Concerned	28.5%	23.0%	−5.6%	−2.3%	−8.8%	0.001 *
Stage of Decision Making	Firmly decided	75.4%	72.5%	−2.8%	−5.9%	0.2%	0.097
VCF Question	Accepting	90.1%	92.0%	1.9%	−0.1%	3.9%	0.096

^a^ Refer to Table 1 for a description of survey items and response grouping. * Statistically significant.

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
