# Peer review of "A Consent Support Resource with Benefits and Harms of Vaccination Does Not Increase Hesitancy in Parents—An Acceptability Study"

_vaccines, 2020, doi:10.3390/vaccines8030500_

Round 1

Reviewer 1 Report

This manuscript is well written and well structured. It addresses a very current topic and adds knowledge to current evidence. It may be of interest to readers.

Author Response

Many thanks for this positive review. 

Reviewer 2 Report

The manuscript by McDonald et al. describes a risk-benefit analysis of providing adequate information to expecting mothers from a largely higher socioeconomic population in Sydney, Australia and studies its effect on parental vaccine hesitancy and their attitude towards the information about the safety and potential side-effects of the vaccines. The authors find encouraging and rather significant changes in the study participants feeling more informed and less concerned about the vaccine side effects and safety. Given the rise in vaccine hesitancy in higher socioeconomic populations across the world and deployment of potential vaccines for the ongoing COVID-19 pandemic on the horizon, this study is very timely and important. The study is well done, manuscript is well written and the sample size is adequate for the drawn conclusions. 

A couple of typos:

Line 273 - "could not to determine" should be "could not determine"

Line 281 - "resources would be ideally be" should be ""resources should ideally be"

Some of the interesting questions, which are beyond the scope of this manuscript, to answer in the follow-up studies are:

  1. How long do the changes in parent's attitudes towards the vaccines last?
  2. It will be great to replicate this study in populations where there is higher vaccine hesitancy or lower coverage e.g. in lower socioeconomic populations.
  3. Given that this was not a strictly randomized sample size, it will be interesting to run the same study in a more randomized population.

Author Response

Thank you for this positive review and for identifying two typographic errors in the manuscript. these have now been corrected.

Reviewer 3 Report

In the study, the authors investigated the parental vaccine hesitancy before and after being given information about vaccination benefits and risks using questionnaires. 416 qualified people were included in the analysis. It showed no hesitancy but more likely to accept vaccination when more informed. It concluded providing information increases decision-making but not increasing hesitancy.

This is a study of high originality to provide first-hand information. However, there are some problems that deteriorated the significance. 

Major problems

  1. Representative samples. The study included a final 416 participants to evaluate their attitude changes after reading vaccine information. However, the authors also admitted only 41.8% finished reading all 8 sections and 21.7% actually "not opening any sections" despite all reporting they had reviewed each section (line 202-204). Clearly, this 21.7% lied and their reports were not reliable. I believe this data should be excluded from the analysis since the conclusion based on that might be misleading. As this 21.7% is a relatively large proportion, it might no longer fit the minimum required sample size of 382 participants after removing them.
  2. Sample size. It was mentioned 34 (8.9%) had concerns (line 207-208). From this information, the total sample size should be 382 participants whose 8.9% is 34 - it is not the claimed 416 participants. What was the real number of participants in the study?

Minor issues

  1. Full names of abbreviations. Please provide the full names when the abbreviations appear for the first time. For example, the SKAI was first used (line 106) but its full name was only given later (line 108). IRSAD was not given the full name in the manuscript text but only appeared in the table.
  2. Inconsistent names, e.g. both PACVSS and PACV-SS were used several times. 
  3. Table 1 grouping/score: the curly braces were overlapped. It's confusing for some of them.
  4. Table 3: what's VCF? Please give the full name in the footnote.

In short, I recommend a major revision based on the above-mentioned reasons.

Author Response

Dear reviewer 3,

Thank you for your review. Please find below a point-by-point response to your comments. We hope your concerns are now adequately addressed. 

Major issues:

Reviewer comment:

  1. Representative samples. The study included a final 416 participants to evaluate their attitude changes after reading vaccine information. However, the authors also admitted only 41.8% finished reading all 8 sections and 21.7% actually "not opening any sections" despite all reporting they had reviewed each section (line 202-204). Clearly, this 21.7% lied and their reports were not reliable. I believe this data should be excluded from the analysis since the conclusion based on that might be misleading. As this 21.7% is a relatively large proportion, it might no longer fit the minimum required sample size of 382 participants after removing them.

Author response:

We have retained the 90 people who did not click on or open the resource because we believe this reflects the 'real world' where parents may be provided with a resource yet not actually read it. We agree with your suggestion of exploring the impact of excluding these parents from the analysis and have now added an additional sub-group analysis to the text (methods lines 183-184 and results lines 240-242) and changes to Table 3. As you'll see there was minimal change in the primary outcome mean scores and no change in the size or direction of the effect. In fact, this sub-group analysis with only the parents who opened some of the resource sections had a greater effect on informed choice and a greater (but not significant) reduction in hesitancy. This sub-group analysis therefore, further strengthens our findings.

Reviewer comments:

2. Sample size. It was mentioned 34 (8.9%) had concerns (line 207-208). From this information, the total sample size should be 382 participants whose 8.9% is 34 - it is not the claimed 416 participants. What was the real number of participants in the study?

Author response:

We realise that there was a typo in the percentage 34/416 is actually 8.2% and this has been corrected. Thank you for drawing this to our attention. We realise also that these results should have actually been in the section with the secondary outcome measures (rather than the intervention fidelity). We think that this might have been overlooked during the refinement of our manuscript. We have moved it to lines 229-233 and believe this makes the findings much clearer. 

Minor issues:

  1. Full names of abbreviations. Please provide the full names when the abbreviations appear for the first time. For example, the SKAI was first used (line 106) but its full name was only given later (line 108). IRSAD was not given the full name in the manuscript text but only appeared in the table.

Author response:

We have corrected these at lines 106 and 277 respectively.

Reviewer comments

2. Inconsistent names, e.g. both PACVSS and PACV-SS were used several times. 

Author response:

This scale is referred to consistently as PACV-SS throughout now. 

Reviewer comments:

3. Table 1 grouping/score: the curly braces were overlapped. It's confusing for some of them.

Author response:

This formatting has been corrected in Table 1

Reviewer comments:

4. Table 3: what's VCF? Please give the full name in the footnote.

Author response:

The footnote has been added as suggested to Table 3. 

Many thanks for your helpful suggestions to improve our manuscript.

Regards

Lyndal Trevena (on behalf of the authors) 

Round 2

Reviewer 3 Report

Congrats~